# In Situ Synthesis of (M:Nb,Ta)C/Ni35 Composite Coating Cladded on 40Cr Steel

**DOI:** 10.3390/ma14237437

**Published:** 2021-12-03

**Authors:** Gaoqiang Jiang, Chengyun Cui, Lu Chen, Yucheng Wu, Xigui Cui

**Affiliations:** School of Mechanical Engineering, Jiangsu University, Zhenjiang 212013, China; 2221903027@stmail.ujs.edu.cn (G.J.); 2112003006@stmail.ujs.edu.cn (L.C.); 2112003004@stmail.ujs.edu.cn (Y.W.); 2211803033@stmail.ujs.edu.cn (X.C.)

**Keywords:** in situ synthesis, laser cladding, composite coating, (M:Nb,Ta)C, 40Cr

## Abstract

To improve the wear and corrosion resistance of the pump barrel material (40Cr steel), a (M:Nb,Ta)C/Ni35 composite cladding coating by in situ synthesis of composite carbides was conducted. The effects of ceramic micro-particles content on the phase composition, microstructure of the coating, structural characteristics of (M:Nb,Ta)C and the tribology and electrochemical corrosion behavior were systematically studied. The increase of ceramic micro-particles changed the morphology of (M:Nb,Ta)C with the size from sub-micron to micron. The (M:Nb,Ta)C dispersed along the grain boundary inhibits the growth of the grains. During friction, the spherical structure exhibited a rolling lubrication effect and the petal structure provided a stronger attachment ability to resist the shear. The corrosion occurred at the grains, exhibiting corrosion pits, in which the high content ceramic micro-particles were relatively shallow. Moreover, a few dot corrosion pits were distributed along the grain boundaries without (M:Nb,Ta)C. Therefore, to improve the corrosion resistance, a thin composite carbide coating with good wear and corrosion resistance was prepared.

## 1. Introduction

The pump barrel material (40Cr steel) in the energy field works for a longer time under severe conditions, such as oil fields, seawater and sand. It is prone to surface failures such as wear and corrosion, which puts forward higher requirements for wear resistance and corrosion resistance. [1,2]. Laser cladding technology can significantly improve the mechanical properties of the parts surface to achieve protection [3,4]. However, the cladding thickness is generally higher than 1 mm, and the dilution rate of the coating and the thermal effect on the thin-walled pump barrel material is high [5]. The parameters were adjusted, and a thin coating (0.5 mm) with a dilution rate of 5.71% was prepared. Another problem is that when conducting performance tests, it was found that the low protective coating limits its all-around performance compared with the traditional coating thickness.

In recent years, many scholars introduced ceramic micro-particles, such as carbides, nitrides and oxides [6,7]. Recently, in order to improve the surface property, composite carbide ceramic micro-particles have been exploited to strengthen the materials [8]. Nam et al. [9] utilized gas tungsten arc cladding to prepare (Ti,W)C-Al_2_O_3_ ceramic reinforced metal matrix composites (MMC). (M:Ti,W)C particles were densely distributed in the MMC, and the hardness was significantly better than that of the single carbide ceramic TiC. Wang et al. [10] studied the Fe-based alloy surface layer reinforced by TiC-VC particles. In this study, the generated TiC-VC particles (2~4 μm) were uniformly dispersed in the matrix, thereby obtaining excellent hardness of the surfacing coating. Gu et al. [11] conducted immersion corrosion tests on TiC-20 vol% SiC and TiC-40 vol% SiC in molten FLiNaK salt at 800 °C;. The corrosion result showed that SiC improved the corrosion resistance of TiC, which in turn improved the corrosion resistance of the entire coating. Nevertheless, the research on (M:Nb,Ta)C composite carbide has not been reported. The atomic sizes of Nb and Ta are very similar (0.1429 nm and 0.1430 nm, respectively), and the crystal structures are also the same (both are FCC structures). Small lattice distortion caused by similar atomic size can improve stability, and the same type of crystal structure enhances solubility. In addition, compared with the external method, the uniform reinforcement phase synthesized in situ is more compatible with the matrix with higher interface bonding strength and thermodynamic stability.

Therefore, the purpose of this work is to improve the wear resistance and corrosion resistance of the thin coating through the introduction of composite carbide ceramic micro-particles with similar atomic sizes and the same crystal structure types. In this paper, the (M:Nb,Ta)C/Ni35 composite coating (about 280~580 μm) was synthesized in situ to reinforce the thin coating cladded on 40Cr steel. The effects of ceramic micro-particle content on the phase composition and microstructure of the composite coating, as well as the structural characteristics and formation mechanism of (M:Nb,Ta)C, were explored systematically. In addition, the tribological mechanism and electrochemical corrosion behavior of the composite coating were described in detail.

## 2. Experimental Procedure

### 2.1. Materials and Methods

The 40Cr steel (150 × 150 × 8 mm) was utilized as a base material, with the chemical composition of C 0.4, Mn 0.6, Si 0.27, Cr 1.1, Cu 0.2, Ni 0.2 and Fe balanced (wt.%). The steel surface was polished, cleaned and blow dried, in succession. The reinforcement layer material was Nb/Ta/graphite/Ni35 alloy mixed powder, and the average particle sizes were 20 μm, 40 μm, 20 μm and 50 μm, respectively. The purity of Nb, Ta and C were all 99.9%, and the chemical composition of Ni35 was C 0.3, B 2.0, Si 3.0, Cr 8.5, Fe 3.0 and Ni balanced (wt.%). 

The process parameters for preparing the composite coating are shown in Table 1. In the Nb–Ta–C/Ni35 mixed powder, the weight percentage of Ta:Nb:C was 1:1:2. Ta–Nb–C and Ni35 respectively increased and decreased by 3% through weight percentage (numbered S1~S6). The preset thickness of S1 was set to 350 μm to prepare a thin coating. Six groups of powders in different proportions were mixed by a planetary ball mill and dried in a drying oven at 60 ℃ for 2 h. Finally, a layer of Ta–Nb–C/Ni35 powder mixed with a small number of adhesive (polyvinyl alcohol glue) was pre-placed on the steel surface by the preset method, and the laser cladding test was performed after 3~5 h at 25 °C. 

IPG-2000 continuous wave laser was employed as cladding equipment, with a maximum output power of: 2000 W, laser wavelength: 1064 nm, spot diameter: 2 mm and focal length: 12 mm. In this experiment, the optimized laser processing parameters were selected based on the first experiment, which includes output power: 600 W, scanning speed: 500 mm/min, overlap ratio between adjacent channels: 50% and argon shielding gas flow rate: 10 L/min [12].

### 2.2. Microstructural Characterization

The (M:Nb,Ta)C/Ni35 composite coating was subjected to wire cutting (12 × 6 × 8 mm) and polishing with sandpaper of 180, 600, 1000, 1500 and 2000 meshes, respectively. Aqua regia (V_HCl_:V_HNO__3_ = 3:1) was employed to perform metallographic corrosion on the polished surface of the samples. X-ray diffractometer (XRD) with copper Kα radiation was used to identify the phase composition, which included scanning range: 30~92° and scanning speed: 1°/min. The optical microscope (OM) and the scanning electron microscope (SEM, Hitachi S-3400N, Hitachi Company, Tokyo, Japan) equipped with an energy dispersive spectrometer (EDS, Hitachi S-3400N, Hitachi Company, Tokyo, Japan) were utilized to observe the microstructure of the samples.

### 2.3. Mechanical Tests

The friction and wear tests were performed on the samples (20 × 20 × 8 mm) by using HT-1000G high-temperature friction and wear tester. The wear test was prepared with a load of 5 N and a speed of 560 r/min, using a GCr15 steel friction ball (Φ5 mm, 60 HRC). The test was performed for 20 min with a wear scar radius of 2 mm. During the test, the wear morphology was observed to understand the wear mechanism.

Through the CHI 660 electrochemical workstation (CHI 660E, Chenhua Company, Shanghai, China), a three-electrode system was utilized for an electrochemical corrosion test of the sample (10 × 10 × 2 mm) at 25 °C. The main parameters include saturated calomel electrode, Pt electrode, working electrode (S0~S6, S0 was the matrix) and neutral 3.5% NaCl electrolyte. The polished 10 × 10 mm coating surface was exposed as the working surface, and the rest was sealed with epoxy resin. Before the experiment, the sample was immersed in NaCl solution for 30 min to obtain a stable open circuit potential (E_OCP_). In the frequency range from 10^5^ Hz to 10^−2^ Hz, an electrochemical impedance spectroscopy (EIS, CHI 660E, Chenhua Company, Shanghai, China) was performed at E_OCP_ with an alternate current signal of 5 mV RMS.

## 3. Results and Discussion

### 3.1. Phase Composition of Composite Coating 

Figure 1 depicts the XRD results of composite coating with different ceramic micro-particle content. The results show that the slender and sharp diffraction peaks of S1 on the crystal planes of (111), (200), (220) and (311) are indexed to Cr_0.19_Fe_0.7_Ni_0.11_, Fe_0.64_Ni_0.36_, (Fe, Ni), and (111), (200) and (220) correspond to Cr, as shown in Figure 1a. When a small number of ceramic micro-particle was added (3 wt.%), the intensity of the diffraction peaks changes slightly, see Figure 1b. The new weak diffraction peaks at 2θ angles of 35.022°, 58.599° and 73.393° show the appearance of (M:Nb,Ta)C phase corresponding to the (111), (220) and (222) crystal planes, indicating that (M:Nb,Ta)C is synthesized in situ. The (M:Nb,Ta)C has a great influence on the microstructure, resulting in the diffraction peaks along the (111), (200), (220) and (311) of the crystal plane, all to be weakened. When adding a large number of ceramic micro-particles (15 wt.%), the intensity of the diffraction peaks obviously changes, see Figure 1c. Comparing Figure 1a,b, the diffraction peaks along the (111), (200), (220) and (311) crystal planes all continue to weaken, indicating that the increase of ceramic micro-particles causes the change of the microstructure. New diffraction peaks appearing at 45.872° and 49.122° are indexed to the compounds Ni_3_C and Fe_3_C, respectively. In addition, the diffraction peaks are enhanced at 35.022°, 58.599° and 73.393°, and new diffraction peaks appear at 40.605°, 69.997° and 87.392°, corresponding to (200), (311) and (400) crystal planes of the (M:Nb,Ta)C phase. It can be inferred that more (M:Nb,Ta)C are synthesized in situ. After the in situ synthesis of (M:Nb,Ta)C, the remaining Nb–Ta combined with Ni to form NbNi and NiTa, and the diffraction peak was observed at 77.923°. In Figure 1b,c, there are gradually reinforced diffraction peaks at 50.668°, which are indexed as Cr_23_C_6_. This is attributed to the “Cr-poor phenomenon”. The Cr in the matrix and powder is easily combined with higher content of graphite to produce chromium carbide and precipitate to the grain boundary. 

Additionally, when comparing Figure 1a–c, the position of the strong peak is observed to shift toward the left, resulting in a corresponding decrease in the 2θ value. On the one hand, according to the Bragg Equation (1) [13]: *2dsinθ* = *nλ*(1)
where *λ* is the wavelength, *n* is the reflection series, *d* is the interplanar crystal distance, *θ* is the angle between the incident ray and the reflected line. The decrease of the *2θ* represents an increase in the *d*. Some alloying elements (such as Nb, Ta, C and Si) and their compounds precipitate, nucleate and grow among the crystal grains during solidification. On the other hand, some alloying elements precipitated too late remain in the crystal lattice, resulting in a slight distortion of the crystal lattice [14].

### 3.2. Microstructure Evolution of Composite Coating

Figure 2a–f show the cross-sectional morphology change of composite coating with different ceramic micro-particle content. There is a good metallurgical bond between the thin coating and the matrix without obvious defects, such as cracks and pores. The thickness of the thin coating without ceramic micro-particles (Figure 2a) was the smallest (only 284 μm), with a relatively flat bonding interface, appearing as the directional solidification and slender columnar dendrites at the bottom of the molten pool. The powder with a smaller particle size has a higher absorption rate to laser energy, causing a smaller matrix melting, a low dilution rate and a smooth bonding interface. Figure 2b,c display that the thickness of composite coating with a small number of ceramic micro-particle increase and the columnar dendrites are slightly smaller.

Further increasing the ceramic micro-particle content (Figure 2d), the thickness of the composite coating and the curvature of the bonding interface significantly increase, presented as a clear wave shape, and the columnar dendrites continue to shrink. Graphite with a larger volume and ceramic micro-particles with a smaller particle sizes have a larger specific surface area and higher nucleation rate, which plays a role in refining the structure. Moreover, it is inhibited by more in situ synthesis enhanced phases, and the large-volume mixed powder absorbs less laser energy to form a higher thickness coating. However, the heat dissipation conditions at the bottom of the molten pool with high-thickness coatings are worse, especially the high-energy area of the Gaussian spot. This allows the temperature to be maintained for a period to delay solidification and melt the matrix, which manifests as a high-curvature bonding interface. When more ceramic micro-particles are added (Figure 2e,f), the coating thickness increases even more (up to 582 μm) with uneven bonding interfaces and the smallest dendrites.

The high-magnification OM diagrams of the bonding interface between the matrix and the coating was taken and observed to have good metallurgical bonding characteristics in Figure 3a–(f1). Additionally, the different areas of S6 in Figure 2f are magnified in Figure 3(f1–f3). The bottom of the molten pool is composed of columnar and equiaxed dendrites. The middle part is mainly columnar, cellular and equiaxed dendrites, while the top part is dominated by columnar and cellular dendrites.

### 3.3. Structural Characteristics of (M:Nb,Ta)C

To explore the change in the structural characteristics of the composite coating with different ceramic micro-particle content, the SEM images are shown in Figure 4. Figure 4a shows the morphology of Ni35 powder cladded on 40Cr steel. The crystal grains grow in an orderly manner with sizes between 3~5 μm, and the grain boundaries are round and smooth. The grain boundaries with a small number of ceramic micro-particles are deformed (Figure 4b,c), and some sub-micron white granular phases are evenly distributed along the grain boundaries without regular geometric shape. The irregular arrangement of atoms at the grain boundary, with defects, such as holes and dislocations, is conducive to the diffusion of the strong carbide-forming elements, Nb–Ta, with a larger atomic radius. Additionally, Nb–Ta can reduce the high energy of grain boundary atoms due to deviation from the equilibrium position, resulting in preferential diffusion to the grain boundary. The density of Nb is equivalent to that of Ni, which plays a role in dispersing and evenly distributing. When the ceramic micro-particle content approaches 9 wt.% (Figure 4d), the particles grow into a spherical shape with a size of about 0.3 μm. In Figure 4e, the grain boundary deformation gradually deepens. The particles gradually grow into micron-level regular geometric shapes, such as quadrilaterals, hexagons, and polyhedrons, causing the abnormal aggregation phenomenon (reverse diffusion). More (M:Nb,Ta)C is synthesized in situ and serves as the core of heterogeneous crystals, which acts as a pin to hinder crystal grain growth and cause grain boundary deformation.

Moreover, Nb–Ta is hard to enter the lattice gap or replace Fe to form a solid solution and distribute it along the grain boundaries. The alloying elements separation at the grain boundaries also reduces the system’s energy [15]. As the ceramic micro-particle content increases (15 wt.%), the cluster particles grow further into petal shapes with sizes of 4 μm and the crystal grains are also refined to 1~2 μm, see Figure 4f. Because convective flow initiates the collision and assembly of particles in the molten state, some particles are engulfed by the continuously advancing solid–liquid interface [16]. Furthermore, the larger atomic size of Nb–Ta preferentially occupies the grain boundary position, which reduces the interface energy and inhibits the grains growth [17].

To further explore the distribution of elements in the composite coating, a EDS line profile on S6 was performed, as shown in Figure 5. The main alloying elements, Fe and Ni, are evenly distributed throughout the composite coating. The content of Fe and Ni where the curve does not pass through the white phase increases and decreases alternately. Since the Fe–Ni alloy undergoes grain boundary segregation during solidification, the solute atom Ni is continuously discharged from the solid phase to the liquid phase, resulting in a large number of Ni in the final solidified grain boundary part. The white phase is identified as Nb–Ta elements, confirming that (M:Nb,Ta)C particles are synthesized in situ.

Moreover, the intensity of the large-size white phase is more extensive, indicating a higher degree of element aggregation. Combined with the point analysis of the white phase in Figure 6, the results show that the content of Nb–Ta and C is high, which again verifies the synthesis of (M:Nb,Ta)C particles. However, it is worth noting that the C content is not the highest.

To explore the overall distribution of the remaining C and Nb–Ta, the area mapping was performed on S6, as shown in Figure 7. Figure 7a,b are the field of view and overlay of S6, respectively, which straightforwardly illustrates the overall distribution of each element. In Figure 7c,d, C and Cr are mainly concentrated in the white phase, followed by grain boundaries, because C with a small atomic radius cannot be entirely dissolved by the strong carbide forming elements, Nb–Ta. According to the atomic diffusion theory, small interstitial solute atoms C easily migrate from one interstitial position of the lattice to another [18]. The loose structure in the grain boundary becomes a channel for the rapid diffusion of C, forming the phenomenon of grain boundary segregation. In addition, the above-mentioned Cr easily follows C distribution to form a “Cr-poor phenomenon”, including the white phase and the precipitation at the grain boundary. In Figure 7e–h, the intensities of Nb–Ta in the petal shape and scattered granular white phases are much higher than that in other regions. In contrast, the intensities of Fe and Ni are extremely low, indicating that they are mainly distributed in areas other than at the white phase. Additionally, referring to the previous report [19,20], the white phase is synthesized in situ; (M:Nb,Ta)C has been further determined.

Figure 8 is schematic diagrams of the cross-sectional structure change and heat dissipation. Viewed from the top of the matrix, the circular laser spot is continuously output to form a multi-melt lap, corresponding to the cross-sectional structure change and the heat dissipation. With the increase of the ceramic micro-particle content, the thickness of the composite coating increases and heat dissipation capacity weakens, causing a gradually increasing curvature of the bonding interface. Additionally, the changes of (M:Nb,Ta)C morphology (irregular granular, spherical, polyhedral and petal shape), size (sub-micron and micron) and number (increased) continuously inhibits the growth of crystal grains. S1 means that it does not contain any composite carbides and the (M:Nb,Ta)C form is a small amount of irregular granular in S2. S4 indicates that the white phase grows into a spherical shape and even a polyhedron in S5. However, some of the white phases appear to be clustered into micron-sized petal shapes in S6.

### 3.4. Formation Mechanism and Morphological Change of (M:Nb,Ta)C

The formation mechanism and morphological changes of (M:Nb,Ta)C particles are summarized. In the molten state, Nb–Ta and C with very different atomic radii tend to form interstitial solid solutions of NbC and TaC. Additionally, Nb–Ta with very similar atomic radii (0.1429 nm and 0.1430 nm, respectively) can replace each other to form (M:Nb,Ta)C substitutional solid solutions. Both of the two components of the FCC stacked crystal structure have significant solubility. When the temperature drops to the melting point of the solid solution, the (M:Nb,Ta)C atom clusters and the attached impurity particles in the liquid phase become crystal embryos of homogeneous nucleation and heterogeneous nucleation. Subsequently, during the random growth of embryos, NbC, TaC and (M:Nb,Ta)C are attached to form a coherent phase boundary with small elastic distortion. During this process, the inter-lattice diffusion of interstitial atoms C and the substitution between adjacent atoms of Nb–Ta occur.

When adding fewer ceramic micro-particles, the (M:Nb,Ta)C grains with less epitaxial layers in the normal direction grow smaller, which are easy to collide and assemble under convection action, appearing as sub-micron irregular granular shapes. With the increase of the ceramic micro-particle, the crystal grains with more epitaxial layers grow to the micron level. Meanwhile, the host elements Fe and Ni have begun to crystallize at the continuously decreasing temperature. Under the obstacle of the grain and the phase boundary, the convective motion slows down, causing the collision to have little impact on the crystal grains. The crystal grains begin to grow into spherical, polyhedral and petal shapes along the direction of the lowest surface energy of the crystal ((111) plane) until the end of solidification [14,21]. 

### 3.5. Friction and Wear Test

Figure 9 depicts the friction coefficient (COF) curves. In general, the COF does not stop increasing at the initial friction stage until the stable stage. However, the entry time and the coefficient value are different. Although the COF of S0 is low, it has been in a slow-rising stage for a long time until 900 s. The wear morphology in Figure 10a shows that there is a wide and discontinuous furrow-like morphology on S0 with a large amount of oxidized flaking debris attached. During severe friction, the hard carbides in the GCr15 steel ball material fall off and remain on the contact surface, causing loose scratches and easy oxidation to produce abrasive wear. The COF of the thin coating (S1) without adding ceramic micro-particles is the largest and the time to stabilize is still very long (about 500 s). In addition, the wear width is also the largest (about 875 μm) and there is a large area of flaking caused by plastic tearing. The wear resistance of the Ni-based thin coating has not improved much, which is mainly attributed to the low thickness of the preset powder.

Additionally, the relative sliding between the GCr15 steel ball material and the parent material causes shear at the contact point, leading to the destruction of the surface film and the occurrence of adhesive wear. When adding fewer ceramic micro-particles (3 wt.% and 6 wt.%), the COF decreases slightly, and the stabilize time continues to shorten (about 400 s). Nevertheless, in the initial stage, the curve of S2 fluctuates the most, indicating that the friction process is the most intense. Moreover, the wear width of S3 is reduced (about 550 μm) and the wear surface is improved significantly. However, there is a protrusion-gathering phenomenon of oxidization at intervals, especially on both sides of the rubbing direction.

On the one hand, the (M:Nb,Ta)C in the parent material inhibits grain growth and achieves fine grain strengthening. On the other hand, loose wear debris of the parent and the GCr15 steel ball material tends to adhere to the surrounding of the exposed (M:Nb,Ta)C of high hardness during the wear process. However, some areas are hard to resist shear stress and are destroyed, forming this kind of interval phenomenon, especially in areas where the shear force is smaller on both sides. When the ceramic micro-particles exceed 9 wt.%, the COF continues to decrease and the stabilize time is reduced to 100~200 s. Inclusively, S4 (oscillating at 0.4325) and S6 (oscillating at 0.4247) have the shortest stabilize time and the smallest COF (lower than S0), which means that the microstructures are more conducive to enhancing the wear resistance of the composite coating. It is worth mentioning that the COF of S5 is relatively larger. The microstructure determines the macroscopic properties. The polyhedral particles in S5 are small and easily fall off, acting as abrasive particles to form higher surface roughness and cause an increase of COF. The shed spherical particles in S4 have a more significant rolling lubrication effect of reducing roughness by grinding convexities and filling concaves. The interval phenomenon of S6 is more evident than others. There is a large area of material accumulation of the parent and the GCr15 steel ball in the outer ring, which is more conducive to preventing further wear of the inner coating. The bare micron-level hard phase can provide stronger attachment and shear resistance, thereby reducing the damaged area under the same shear stress. However, the accumulated material of the parent and the GCr15 steel ball in the outer ring is hard to be damaged by small shear stress and is left behind.

### 3.6. Electrochemical Corrosion Test 

Figure 11 describes the potentiodynamic polarization curves and the Nyquist plots of S0 and S3~S6, and the test results are attached in Table 2. The self-corrosion potential (Ecorr) of S0 (−0.522 V) is smaller than that of S3~S6 (−0.486~−0.463 V), testifying that the corrosion resistance has been improved significantly. In addition, the self-corrosion potential does not show a significant change with the increase of ceramic micro-particle contents. However, the current density of the composite coating increases first and then decreases. When the content exceeds 12 wt.% (S5 and S6), the polarization resistance reaches the maximum (3093 Ω and 3559 Ω, respectively), and the current density reaches a minimum (1.039 × 10^−5^ A·cm^−2^ and 1.065 × 10^−5^ A·cm^−2^, respectively), which implies that they have the lowest corrosion rate and the most substantial corrosion resistance. This is consistent with the calculation of the corrosion rate in Table 2.

In Figure 11b, the arc radii of S3~S6 are larger than that of S0, and the arc radius of the sample with higher ceramic micro-particle content shows a decreasing trend after increasing. Similarly, the arc radii of S5 and S6 are relatively larger (600 Ω·cm^2^ and 700 Ω·cm^2^, respectively), which is consistent with the potentiodynamic polarization curve. Besides, although the arc radius of S4 is the largest, the current density is relatively larger, illustrating that the corrosion resistance is not the best.

To further explore the corrosion morphology, the SEM images are exhibited in Figure 12. S0 is severely corroded and appears with deep groove-shaped corrosion pits in Figure 12a. Corrosion occurs along the boundary of the flaky tissue, resulting in a flaky pit wall. Figure 12b–f show that the corrosion occurs at the grains, appearing in different sizes and depths of grain corrosion pits. The electronegativity difference between matrix and coating causes a forming of a small primary cell; therefore, corrosion mainly occurs along the depth direction perpendicular to the coating surface [22]. In addition, with the increase of ceramic micro-particle content, the size and depth of the grain corrosion pits gradually become smaller. This is attributable to the in situ synthesis of more (M:Nb,Ta)C and the continuously precipitated Cr element at the grain boundary, which broadens the grain boundary width and restricts the crystal grains growth, causing the porosity to continuously decrease. For S5 and S6 in particular, as shown in Figure 13, the porosity reaches the minimum (39.89% and 22.08%, respectively), which indirectly indicates the enhanced ability to resist corrosion. Meanwhile, the contact area between the crystal grains and the electrolyte, as well as the corrosion depth, is also effectively reduced to prevent further corrosion. 

In Figure 12e,f, EDS element concentration analysis is performed on selected areas A~D on the corrosion surface, and the results are shown in Table 3. The white phase at point A shows the presence of (M:Nb,Ta)C at the grain boundaries without being corroded, accompanied by oxidation. The corrosion grain pits at the black phase of point B also have an oxide layer, and there is also an increase in C content. The C content decreases in the grain boundary of point C. As mentioned in the previous section, C is mainly concentrated in grain boundaries and composite carbides rather than crystal grains. However, there is an abnormal phenomenon that the C content increases at the grain pits at point B. The essence of electrochemical corrosion is a process in which metals are oxidized through a pair of conjugated oxidation–reduction reactions. The Fe atoms in the grain pits are corroded and oxidized, causing the solid solution C atoms dissolved in the electrolyte to be reduced.

Furthermore, a small number of corrosion pits in the dot-shape (point D) without Nb–Ta elements are observed along the grain boundary, and the C content is also the lowest. Specifically, the corrosion pits are distributed along the grain boundaries without (M:Nb,Ta)C, which indirectly certificates the effect of (M:Nb,Ta)C on the corrosion resistance of the composite coating. At point E, a large petal shape (M:Nb,Ta)C containing high Nb–Ta restrains the grain corrosion and presents the shallowest grain corrosion pits. In addition, the Nb–Ta element-formed oxides force the reduction and increase of the Cr dissolved in the electrolyte [22].

## 4. Conclusions

In this paper, the (M:Nb,Ta)C/Ni35 composite coating prepared by the in situ synthesis was discussed. The increase of ceramic micro-particles changed the morphology of (M:Nb,Ta)C constantly with the size from sub-micron to the micron. The (M:Nb,Ta)C dispersed along the grain boundary inhibits the growth of grains. During friction, the spherical structure exhibited a rolling lubrication effect and the petal structure provided a stronger attachment ability to resist shear. The grain corrosion pits with a high content of ceramic particles are relatively shallow. Some corrosion pits in the dot-shape were distributed along the grain boundaries without (M:Nb,Ta)C.

## Figures and Tables

**Figure 1 materials-14-07437-f001:**
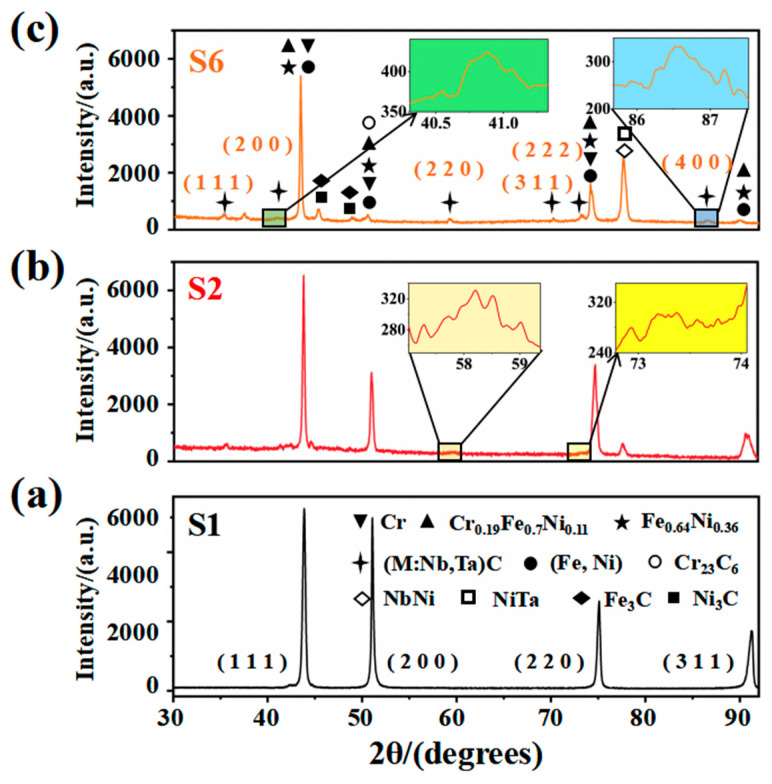
XRD results of composite coating with different ceramic micro-particle content (**a**) S1; (**b**) S2; (**c**) S6.

**Figure 2 materials-14-07437-f002:**
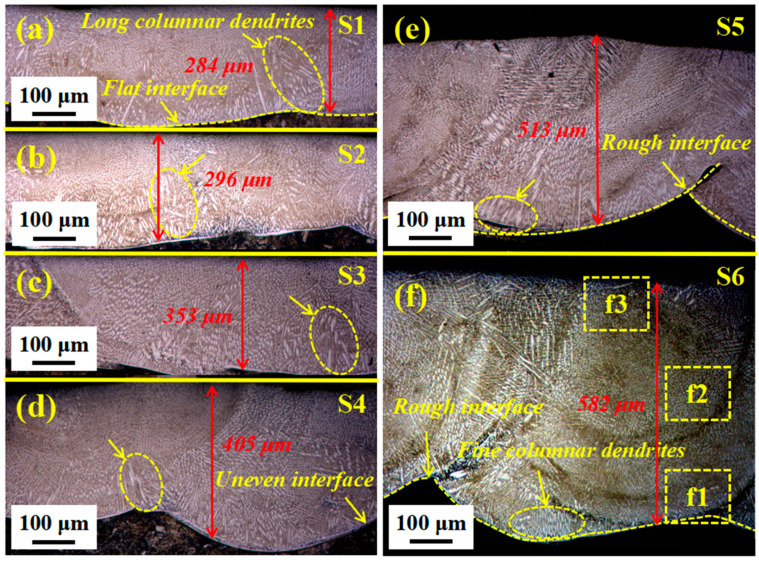
Typical OM diagram of a cross-section of composite coating with different ceramic micro-particle content. (**a**) S1; (**b**) S2; (**c**) S3; (**d**) S4; (**e**) S5; (**f**) S6.

**Figure 3 materials-14-07437-f003:**
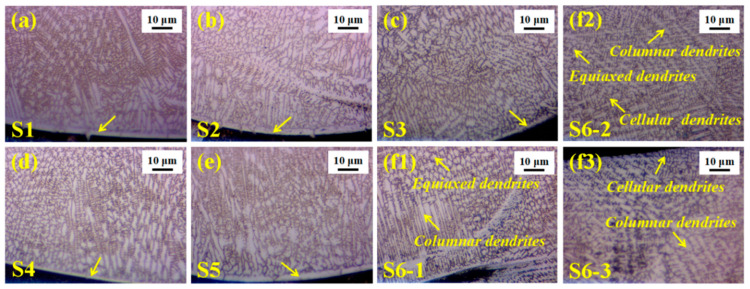
The high-magnification OM diagrams of the bonding interface and the different areas of S6. (**a**) S1; (**b**) S2; (**c**) S3; (**d**) S4; (**e**) S5; (**f1**) S6 (the bottom); (**f2**) the middle; (**f3**) the top.

**Figure 4 materials-14-07437-f004:**
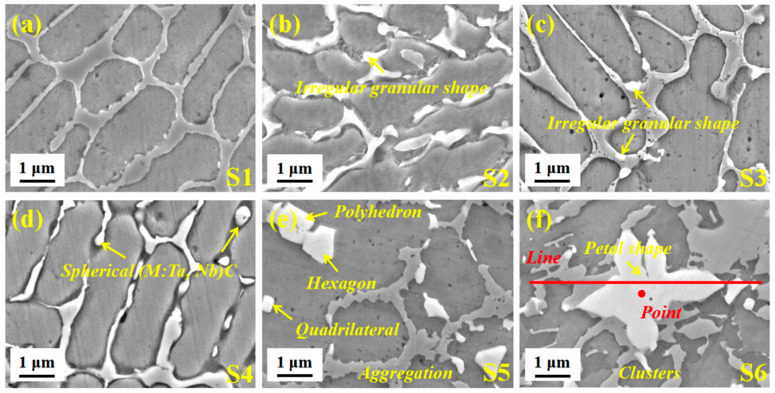
Typical SEM images of composite coating with different ceramic micro-particle content. (**a**) S1; (**b**) S2; (**c**) S3; (**d**) S4; (**e**) S5; (**f**) S6.

**Figure 5 materials-14-07437-f005:**
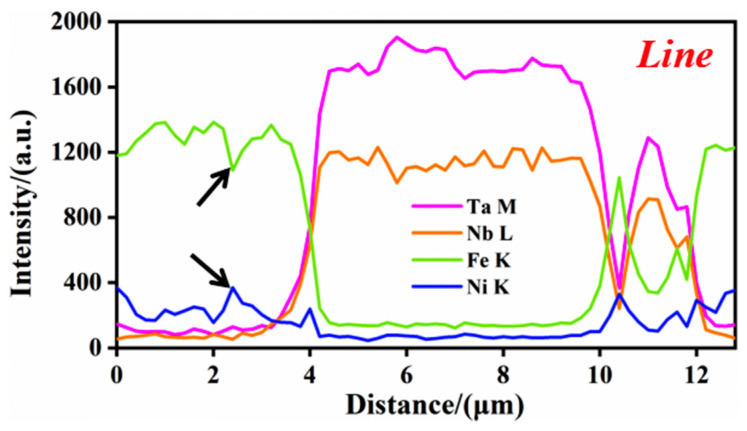
The EDS line profile of element concentration in the typical area in Figure 4.

**Figure 6 materials-14-07437-f006:**
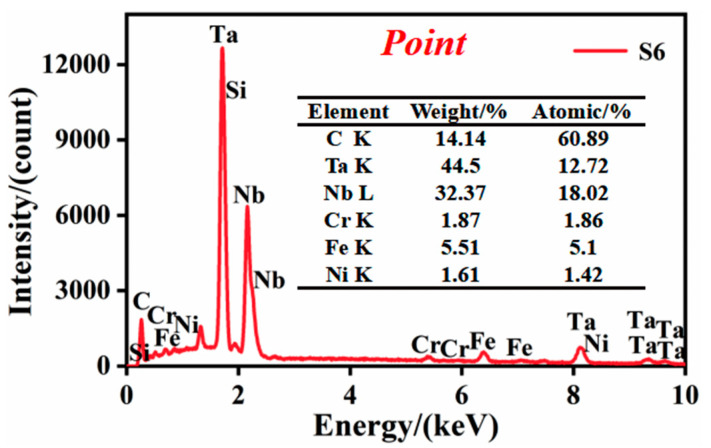
The point analysis of element concentration of the point in Figure 4.

**Figure 7 materials-14-07437-f007:**
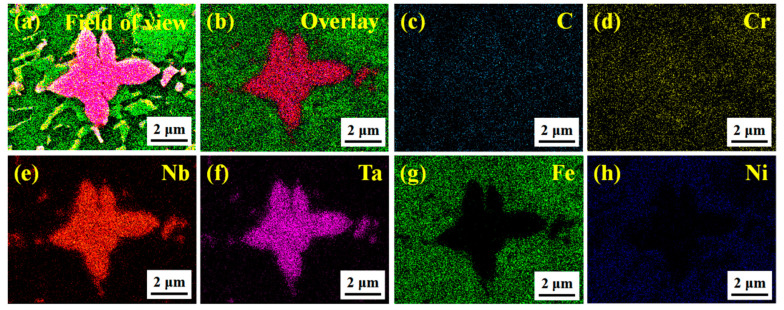
Typical area mapping of S6 in Figure 4. (**a**) Field of view; (**b**) overlay; (**c**) C; (**d**) Cr; (**e**) Nb; (**f**) Ta; (**g**) Fe; (**h**) Ni.

**Figure 8 materials-14-07437-f008:**
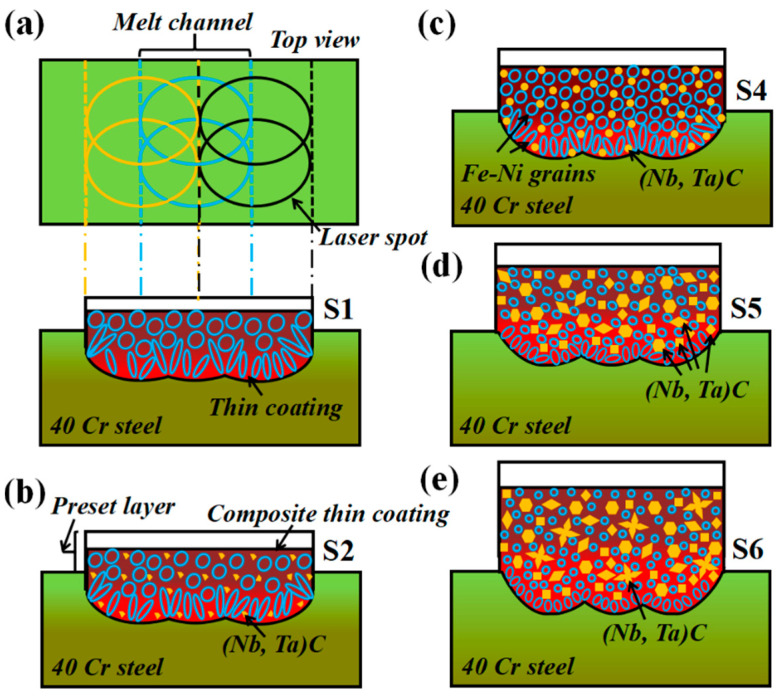
Schematic diagrams of cross-sectional structure change and heat dissipation of composite coating with different ceramic micro-particles. (**a**) S1; (**b**) S2; (**c**) S4; (**d**) S5; (**e**) S6.

**Figure 9 materials-14-07437-f009:**
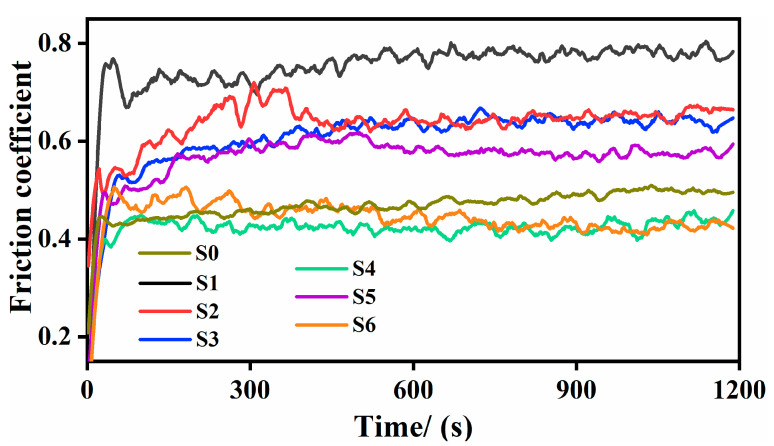
Friction coefficient curves of composite coating with different ceramic micro-particles.

**Figure 10 materials-14-07437-f010:**
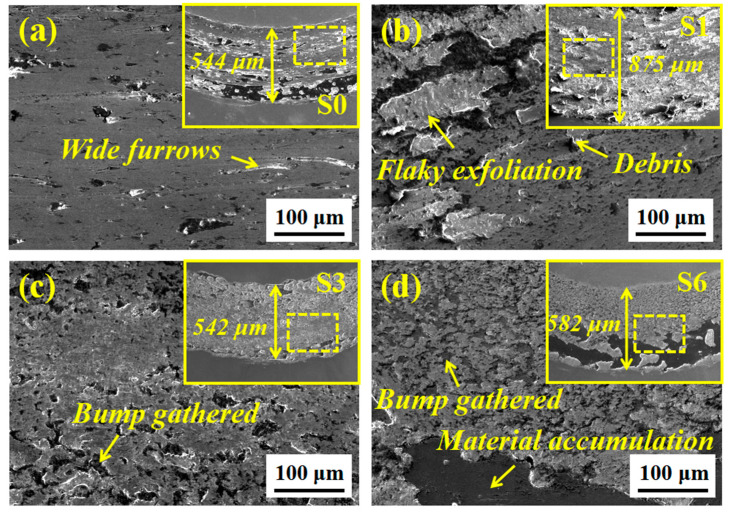
Typical SEM diagrams of the wear morphology of composite coating with different ceramic micro-particle. (**a**) S0; (**b**) S1; (**c**) S3; (**d**) S6.

**Figure 11 materials-14-07437-f011:**
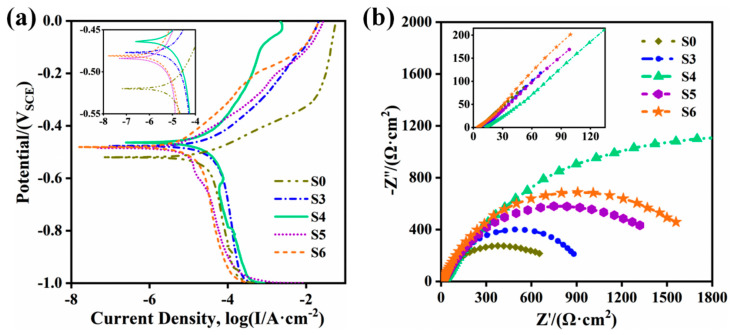
Potentiodynamic polarization curves and Nyquist plots of composite coating with different ceramic micro-particles. (**a**) Potentiodynamic polarization curves; (**b**) Nyquist plots.

**Figure 12 materials-14-07437-f012:**
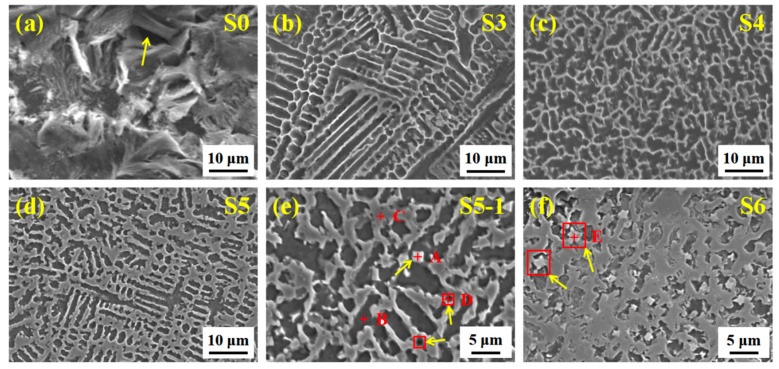
Corrosion surface morphology in Figure 11. (**a**) S0; (**b**) S3; (**c**) S4; (**d**) S5; (**e**) enlarged image of S5; (**f**) S6.

**Figure 13 materials-14-07437-f013:**
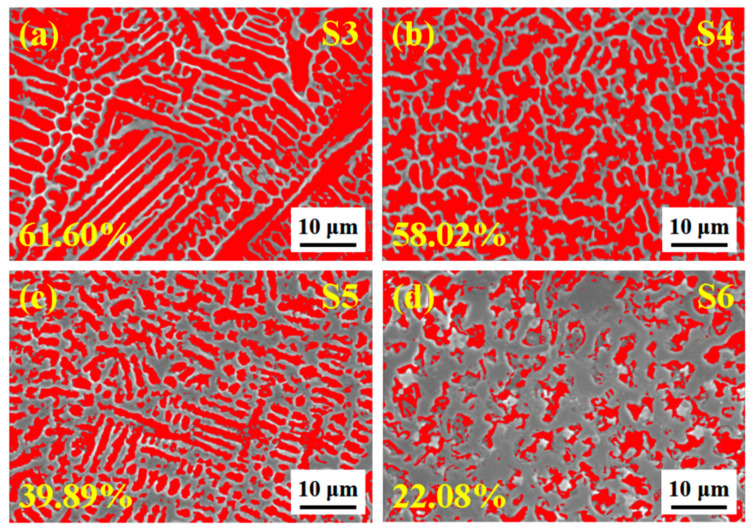
Quantitative analysis image and porosity of corrosion area in Figure 12. (**a**) S3; (**b**) S4; (**c**) S5; (**d**) S6.

**Table 1 materials-14-07437-t001:** Processing parameters of composite coating with different ceramic micro-particles.

No.	Nb:Ta:C (wt.%)	Nb–Ta–C (wt.%)	Ni35 (wt.%)
S1	0	0	100
S2	1:1:2	3	97
S3	1:1:2	6	94
S4	1:1:2	9	91
S5	1:1:2	12	88
S6	1:1:2	15	85

**Table 2 materials-14-07437-t002:** Potentiodynamic polarization test results in Figure 11.

No.	E_corr_/V	I_corr_/A	R_p_/Ω·cm^2^	V_corr_/A·cm^−^^2^
S0	−0.522	2.673 × 10^−^^5^	1010	0.932 × 10^−^^3^
S3	−0.474	3.527 × 10^−^^5^	1028	1.036 × 10^−^^3^
S4	−0.463	3.676 × 10^−^^5^	1289	0.722 × 10^−^^3^
S5	−0.486	1.039 × 10^−^^5^	3093	0.296 × 10^−^^3^
S6	−0.479	1.065 × 10^−^^5^	3559	0.286 × 10^−^^3^

**Table 3 materials-14-07437-t003:** Elemental concentrations of the corrosion areas in Figure 12 (wt.%).

Area	Fe	Ni	Nb	Ta	C	Cr	O
A	13.49	3.57	29.82	33.02	13.97	1.12	5.01
B	60.01	21.34	0.09	0.09	12.63	1.00	4.26
C	72.26	17.15	1.46	0.54	5.82	1.64	1.13
D	71.42	24.05	0	0	2.11	1.31	1.01
E	6.42	1.68	38.6	39.38	11.58	2.34	/

## Data Availability

We choose to exclude this statement.

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
