# Peer review of "In Situ Synthesis of (M:Nb,Ta)C/Ni35 Composite Coating Cladded on 40Cr Steel"

_materials, 2021, doi:10.3390/ma14237437_

Round 1

Reviewer 1 Report

The manuscript presents the microstructural characteristics, wear and correction studies on (M: Nb, Ta) C/Ni35) cladded on 40Cr steel. This manuscript at its present form not suitable for publication. This manuscript need extensive rewriting for its acceptance. The following suggestion/corrections are provided for the improvement of the manuscript.

  1. The title may be modified as ”In-situ synthesis of (M:Nb,Ta)C/Ni35 composite coating cladded on 40Cr steel”
  2. In several places the authors mentioned ‘to reinforce thin coating cladded on’...This dos not convey a proper meaning. What is the meaning of ‘reinforce’ in the sentence?
  3. ‘The results demonstrated that with the increase of ceramic micro-particles content, the changes of (M: Nb, Ta) C morphology, size and number…” in this sentence, what does number refer to?
  4. First two lines in the introduction section is incomplete. Rewriting is needed
  5. Cladding itself a process of depositing a layer of material over a substrate. Why author has mentioned ‘thin coating cladding’ in the introduction section?
  6. Low protective coating- what is the meaning?
  7. Within 5 lines, the authors have used 9 citations. Unnecessary citations may be removed or each reference must be discussed separately.
  8. ‘Fe-based coating, using content changes..” The sentence does not convey the proper meaning.
  9. ‘our purpose was to further reinforce the mechanical properties…” The mechanical properties cannot be reinforced…. Proper modifications needed in the paragraph.
  10. 1 section may be renamed as ‘ Materials and Methods’
  11. Table 2 may be removed.
  12. The authors presented and discussed the XRD of S1, S2 and S6 specimens. Why S3,S4 and S5 specimens are discussed. In fig.1 b,c images are not properly indexed.
  13. The word ‘calibrated’ may be replaced as ’ indexed’
  14. What is ‘Cr-poor phenomenon’ – A proper explanation is needed or a suitable citation.
  15. How metallurgical bond characteristics are ensured only with low magnification OM images?
  16. The porosity quantification is missing.
  17. The EDS spectrum is presented only for S6...Why?
  18. The quantitative analysis of wear is missing. In any wear studies, the parent and the ball materials properties must be stated.
  19. Conclusions must be presented with more specific to your finding.

Reviewer 2 Report

Find the attached document for Author

Round 2

Reviewer 1 Report

Minor corrections are suggested . The corrected manuscript is attached
